# Access Control, Key Management, and Trust for Emerging Wireless Body Area Networks

**DOI:** 10.3390/s23249856

**Published:** 2023-12-15

**Authors:** Ahmad Salehi Shahraki, Hagen Lauer, Marthie Grobler, Amin Sakzad, Carsten Rudolph

**Affiliations:** 1Department of Computer Science and Information Technology, La Trobe University, Melbourne 3086, Australia; 2Department of Mathematics, Natural Sciences, and Computer Science, Technische Hochschule Mittelhessen, 35390 Gießen, Germany; hagen.lauer@mni.thm.de; 3Cybersecurity and Quantum Systems (CQS), CSIRO’s Data61, Melbourne 3168, Australia; marthie.grobler@data61.csiro.au; 4Dep of Software Systems & Cybersecurity, Monash University, Melbourne 3800, Australia; amin.sakzad@monash.edu (A.S.); carsten.rudolph@monash.edu (C.R.)

**Keywords:** access control, healthcare system, key management, privacy, security, trust, Wireless Body Area Networks (WBANs)

## Abstract

Wireless Body Area Networks (WBANs) are an emerging industrial technology for monitoring physiological data. These networks employ medical wearable and implanted biomedical sensors aimed at improving quality of life by providing body-oriented services through a variety of industrial sensing gadgets. The sensors collect vital data from the body and forward this information to other nodes for further services using short-range wireless communication technology. In this paper, we provide a multi-aspect review of recent advancements made in this field pertaining to cross-domain security, privacy, and trust issues. The aim is to present an overall review of WBAN research and projects based on applications, devices, and communication architecture. We examine current issues and challenges with WBAN communications and technologies, with the aim of providing insights for a future vision of remote healthcare systems. We specifically address the potential and shortcomings of various Wireless Body Area Network (WBAN) architectures and communication schemes that are proposed to maintain security, privacy, and trust within digital healthcare systems. Although current solutions and schemes aim to provide some level of security, several serious challenges remain that need to be understood and addressed. Our aim is to suggest future research directions for establishing best practices in protecting healthcare data. This includes monitoring, access control, key management, and trust management. The distinguishing feature of this survey is the combination of our review with a critical perspective on the future of WBANs.

## 1. Introduction

Electronic-Healthcare (E-healthcare) has revolutionised the healthcare system, with Wireless Body Area Networks (WBANs) being an integral part of it [1]. By using e-healthcare technology, a patient’s personal data can be accessed remotely in real-time, from any location. These data are shared among different users of the e-healthcare system to be used by stakeholders with appropriate permission levels to access the medical resources. Since the validity and reliability of healthcare data against a variety of threats (e.g., eavesdropping or data manipulation) is a high priority that can be achieved by addressing privacy and security in WBANs, researchers are particularly motivated to design protocols that are reliable and efficient [2,3,4,5,6].

Applications for WBANs include healthcare, emergency services, military, education, professional sport and fitness, consumer electronics, and games. In this survey, we focus on healthcare applications that allow radio communication to and from sensors, aiming to capture physiological data and monitor vital parameters, capture body activity, and help to improve quality of life [7,8,9].

When considering WBAN security and privacy, it is important to recognise how data flow and how they are transferred between medical devices and stakeholders, based on their particular roles and responsibilities. WBANs can communicate with each other if needed for data dissemination, for example, via WiFi, ZigBee, and Narrowband [7,10,11]. In general, WBANs include several types of emerging sensing devices, including medical wearable and implanted sensors used in Electrocardiography (ECG), Electroencephalography (EEG), Electromyography (EMG), and blood pressure and motion detectors [12,13,14,15,16,17]. They may also include a Control Unit (CU) (e.g, a patient’s smartphone) [18,19] that collects physiological and highly sensitive healthcare data from connected sensors and forwards this information to other devices using short-range communication technology [12,20]. The CU may also act as a gateway between the sensors and a medical server monitored by a healthcare service provider (such as a doctor and nurse) [21,22]. Wearables and sensors are primarily designed and configured to monitor physiological data to support various e-health applications (see Figure 1) [23,24].

In a typical healthcare scenario, it is important to know who has access to what information, how they access it, and for what reason. For example, a General Practitioner (GP) may have full access to medical data in a medical server or CU carried by the patient, whereas a pharmacist filling a patient’s prescription may only have access related to that specific prescription [25]. Collecting sensitive data from a patient in a mobile environment and subsequently sharing this information on interconnected WBANs presents numerous security, privacy, and trust issues. To tackle these challenges, the present study conducts an in-depth exploration of WBANs in healthcare contexts. Prior extensive research and remedies addressing these challenges are inadequate as advancements in WBAN security, privacy, and trust are still evolving.

### 1.1. Why This Paper Is Important

In this paper, we survey several recent works and present an overview of access control, key management, and trust in WBANs. We briefly summarise and review related surveys and other review publications in the domain of WBANs over eight years, considering papers published from 2015. We include healthcare-specific studies, as well as works on other areas related to WBAN and medical devices, to demonstrate the importance of this research and identify missing topics and gaps in the existing literature as well as the advantages and disadvantages of published surveys and review papers.

Table 1 shows a selection of publications based on different forms of communications in WBANs. In [3], the most relevant Quality of Service (QoS)-based routing schemes in WBANs are classified. In [20], the current healthcare sensor and research prototypes are categorised to provide the best solutions for inter-communications and conclude with some future research about the wearable market. In [23], the healthcare application and different types of communication and their standards are investigated, as well as WBAN technologies. Ref. [26] investigated the future of Implanted Medical Devices (IMDs) and discussed future directions to protecting such medical devices. Ref. [27] examined communication technologies and introduced future work for channels, data, bandwidth, power consumption, and mobility. In [28], wireless body sensor networks (WBSNs) are investigated, including relevant routing and energy protocols and methods applicable to such devices and scenarios. In [29], the correct communication architecture related to WBANs is investigated, including key management challenges and issues to mitigate the problem of privacy and integrity of aggregated data. Ref. [30] surveyed the WBAN communication architecture and related security and privacy matters and explored future challenges of WBAN communication. In [31], current communication technologies in WBAN systems used in healthcare domains are investigated, including home infrastructure. Ref. [32] focused on more general matters of communication in electronic healthcare systems.

The article also outlined future challenges in the adoption of Software-Defined Networking, Energy Harvesting, and Blockchain technologies in the field. Lastly, state-of-the-art routing schemes in WBSNs are surveyed based on recent standards and publications in [33]. Ref. [34] investigated physiological and other signals for a key exchange scheme on a sensor in a Body Area Network (BAN). The authors revealed that there is no unique standard for measuring such a scheme’s efficiency. Ref. [35] surveyed WBAN architectures and communication technologies based on the IEEE 802.15.6 standard. The authors of [36] presented a survey on healthcare data security in WBANs to identify various attacks with time, cost, and less memory consumption. Ref. [37] reviewed relay-based communications (RBC) in a WBAN and deeply investigate the various techniques to mitigate problems related to interference in non-cooperative networks using the concept of RBC and classify QoS-based routing schemes in WBANs. The authors of [38] demonstrate the general idea of WBAN technologies using different scenarios. The authors revealed that quantum technology is a promising technology that will use in future technology to generate keys within medical sensors in WBANs. The authors of [39] reviewed WBAN applications and technologies that cover security, antennas, Energy Harvesting, and power management.

Despite the number of useful surveys and review papers in the area of WBAN [3,20,23,26,27,28,29,30,31,32,33,34,35,36,37,38,39], the need remains for a survey with a strong focus on security, privacy, and trust. Such a survey should include an investigation on WBAN background and architectures, as well as a general system model to help identify and present a relevant summary of specific future directions and open issues, including access control, key management, and trust, to meet the requirements required by the WBAN application.

### 1.2. Our Contribution

To address this need, we investigate and recognise the challenges and open issues in WBAN communication; doing this is crucial when evaluating the performance of the developed model in terms of security, privacy, and trust implications. Therefore, the main aim of this paper is to provide a detailed understanding of healthcare and WBANs and explore the current open issues and challenges based on intra-, inter-, and beyond-WBAN communication. We highlight the significance of this paper by comparing our work to selected high-quality review and survey papers (refer to Table 1). This work can be used as a solid building block towards the design of a WBAN system. The Abbreviations section shows the commonly used abbreviations in this paper. Briefly, the contributions of this paper are to:conduct a literature review on research carried out in WBANs and related technologies in the healthcare system (Section 2).categorise WBAN applications and communications based on real healthcare scenarios (Section 3).discuss open issues and provide future recommendations and opportunities and reliable solutions for developing WBAN and e-healthcare applications in terms of access control, key management, and trust (Section 4 and Section 5).

The paper is organised as follows: Section 2 discuss the background of WBANs. Section 3 presents a generic healthcare model, with a discussion on current open research issues on access control, key management, and trust in Section 4 and Section 5. Finally, Section 6 concludes the study and presents future work.

## 2. Background on WBANs

A WBAN is a type of body area network and an emerging technology of wearable computing devices, aimed at improving the quality of healthcare services and improving quality of life [40]. A WBAN can be defined as a wireless network of heterogeneous wearable computing devices that is used for the continuous remote monitoring of physiological data in a medical environment [41]. Our paper is mainly concerned with WBANs in the healthcare environment, which typically include two types of devices: medical sensors (e.g., ECG) and a CU (e.g., smartphone). The medical device industry plays an important role in the healthcare environment as these devices are configured to capture physiological data and forward this sensitive information to the CU, which in turn works as a gateway between the sensors and the access point to broadcast related data to the medical server (e.g., cloud). Figure 2 shows a WBAN with sensors in industrial applications placed at various locations on the body to forward sensed data to a medical server through a more computationally powerful device such as a smartphone or other personal device. Healthcare service providers (e.g., a doctor) can then access the medical data remotely or physically monitor the state of patients in real-time [42]. We explore WBANs based on the type of sensor nodes involved and their different applications [43].

### 2.1. Edge Computing in WBANs

Figure 3 presents the basic edge computing architecture. This is comparable to other industrial three-layer edge computing architectures [44]. We describe the WBAN architecture in terms of layers for connectivity and sensors (i), network (ii), and cloud and applications (iii).

The first layer is referred to as a *connectivity or sensors layer*. Typically, this layer encompasses different kinds of Internet of Things (IoT) devices [45,46]. In a WBAN system, it can be used to describe different medical sensors (implant and wearable) at the leafs of a network and CUs as the nodes that join them into the next layer.

In an edge scenario, the *networklayer* describes what is referred to as an edge device. It receives healthcare data generated by the sensor layer and specifies the pathways to transfer the processed data to the upper layer. Additionally, this layers includes several devices, such as a router, an insider/outsider antenna, and a smartphone. Each of these devices uses different communication technologies [47], such as RFID and WiFi. Software-Defined Networking and Network Function Virtualization are well-know technologies that have been developed for this layer [48,49].

The third layer in the edge computing architecture is the extitcloud or application layer, populated by various providers and services [50,51,52]. In an edge computing scenario, this layer is both a producer and consumer of data, processing the data along the edges in the first two layers. Figure 2 indicates that the third layer interfaces with the second layer, exchanging and contextualizing information. Classically, the cloud and application layer provides different types of databases to store and further process healthcare data. This is becoming more relevant despite the apparent security and privacy concerns; consider the example of artificial intelligent reasoning. We observe that machines rapidly match and outpace humans in the detection and prediction of diseases based on data alone. As the amount and quality of data increases, we envisage an inherent need to combine the current computing power in this layer with access to relevant healthcare records in the near future [53,54].

### 2.2. Nodes Used in WBANs

A node can perform basic processing, collect data, and communicate with other nodes in small- or large-scale networks. We classify nodes in terms of their functionality:-**Implant Sensor:** This sensor is planted in or under a patient’s skin to monitor vital data. An implant node in a WBAN is called an implantable medical device (IMD) and is programmed/configured to sense specific data [55]. An external device is used to program the IMD using a wireless interface to send and receive data. The size and location of an IMD in a WBAN are critical factors because of the energy and storage capacity that are required in the healthcare system [56].-**Wearable Sensor:** This sensor can be attached to the body [57]. Sensors are configured to sense physiological data from the body and transfer the data to a CU through a wireless medium. These devices allow one to check vital data from the body at any time [20].-**Control Unit (CU):** A CU (e.g., a smartphone) collects related data from the sensors and forwards it to a medical server. It works as a gateway between a sensor attached in or on the body and any external devices and therefore must have a good battery and memory. Smartphones and other personal devices are examples of CUs that are used in WBANs.-**Other Node:** This type of node is in the vicinity of the human body, such as an access point or a computer connected to the internet. This node transfers data between the CU and healthcare service providers. The advantages of this over an internal node are additional computing power and storage.

### 2.3. Sensor Node Hardware

In this section, we highlight and describe some of the existing sensor types used in WBANs. Specifically, we look at blood pressure [58], Electrocardiogram (ECG) [59], accelerometer [59], electromyogram [14], carbon dioxide (${\mathrm{CO}}_{2}$), Electrocardiography (EEG) [14], blood glucose [58], temperature, and the atmospheric moisture sensor hardware used in WBANs. Some sensors commonly used in WBANs are listed, and their features are compared in Table 2. Other medical sensing devices have also appeared on the market [60]. These are based on 5G technologies, such as Sensium Digital Plaster, Sensium Life Pebble, Fitbit, Apple iPhone (using Health application), Sensor Strip, and Libelium [61].

-**Blood pressure sensor:** this is a non-invasive sensor that measures diastolic and systolic blood pressure, two of the principal vital signs in the human body [58].-**Electrocardiogram (ECG) sensor:** the ECG records heart activity and directs the signals to a medical server for monitoring by a physician. In order to monitor these signals, a number of ECG are attached to the skin.-**Accelerometer sensor:** this sensor helps physicians to monitor the patients (e.g., crawling, running) [59].-**Electromyogram (EMG) sensor:** this is a sensor for neuromuscular monitoring while the patient is at rest. EMG is a useful sensor to avoid post-operative residual curarization [59].-**Carbondioxide (**${\mathrm{CO}}_{2}$**) sensor:** this sensor calculates gaseous ${\mathrm{CO}}_{2}$ levels to monitor oxygen concentration.-**Electrocardiography (EEG) sensor:** this sensor captures brain activity. Data are sensed and redirected to an amplifier for processing [59].-**Blood glucose:** this sensor monitors the amount of blood glucose in the human body [58].-**Temperature sensor:** this sensor is used to calculate the temperature of the human body and environment.-**Atmospheric moisture sensor:** this sensor is used to calculate the humidity of different environments.

There are several cryptographic schemes employed in WBAN as a secure channel where the communication is unicast from system to sensors and vice versa. The main idea is to generate the key pairs based on the future of wireless channels. To generate a strong secret key, the bit-rate needs to be carefully considered. Increasing the bit-rate would result in a dramatic increase in the key size as well. This brings several challenges in terms of key management and power consumption in practice. Additionally, latency and power consumption are considered as key requirements in WBAN applications as increasing battery lifetime at the cost of higher latency may be necessary in WBANs [62,63].

### 2.4. WBAN Applications

In this section, we highlight and describe the existing WBAN application types used in different environments, such as healthcare. WBAN applications allow for a radio connection to sensors and devices for monitoring aspects such as heart rate [64]. The applications of WBANs can be divided into three categories: (1) in-body applications, which include implant devices; (2) on-body medical applications, which include wearable sensors [20]; and (3) on-body non-medical applications, which include entertainment devices. These sensors monitor the state of the body and transmit data to medical servers to be used by any healthcare service provider [65]. WBAN applications in medical, sport, and military environments are detailed next.

*Medicalapplications* in WBANs can be divided into three categories.

-**Monitoring of human physical data:** Some types of sensors collect physical data from the human body and send the data for further services to healthcare providers, such as hospitals or insurance agencies [66].-**Tracking and monitoring doctors and patients in a hospital:** Patients and doctors carry specific sensors; each sensor has a specific function. For example, a sensor node may monitor the blood pressure of the patient, while another sensor may monitor body temperature. Sensors carried by healthcare professionals in the hospital may enable them to track or locate, as well as direct them towards, specific patients [66].-**Drug administration in hospital:** A major concern in hospitals is that a patient receives the wrong medication [66]. With the introduction of drug sensors, doctors can reduce medication errors and avoid other problems caused as a consequence of this. For example, these sensors can detect and check for sensitivities and allergies to drugs [66].

In *sport applications*, sensors are placed on or around the body to monitor fitness, posture, and movement. These sensors can detect the speed and position of the body, as well as other important vital signs (such as heartbeat and body temperature) [67].

In *military applications*, WBANs are used in the transmission and use of military information. This application may increase the performance of a soldier in individual operation. The sensors, in-body or on-body, monitor vital information about the soldier and supply key information about the environment, position, and posture of the soldier to aid in avoiding threats [68].

## 3. General Healthcare Communication, Standards, and Technology

In this section, we present a general healthcare scenario based on a data model describing the healthcare system. We describe the overall WBAN system architecture and present the related communication technologies and standards based on WBAN system architecture.

### 3.1. General Healthcare Scenario

As stated above, relevant data from patients are transmitted from sensors via the WBAN to a medical database server to be recorded and used by different stakeholders and users. Stakeholders and users can access the recorded data on both the medical server and CU, or request new services remotely from any place. Thus, the privacy of the related personal data is very important and needs appropriate management. It is necessary to provide privacy for a patient’s personal data both during transmission and when stored on remote servers to protect critical data and prevent unauthorised access. To do this, we first identify roles and policies related to users.

In a healthcare scenario, we assume a patient (the subject) has a condition (e.g., high heart rate) that manifests during his/her travel to another location. Three scenarios can be considered here. Firstly, under *normal circumstances* a professional staff member (e.g., doctor) is remotely linked to a smartphone and requests the patient’s information. After reading the data requested from the CU, the healthcare service provider may provide this information to the health professional. Secondly, in an *emergency situation*, a doctor connects remotely or directly to the sensors on the patient’s body and reads the medical profile to monitor critical data in real-time. Thirdly, hospital staff need to obtain access to the patient’s WBAN if the patient undergoes *treatment* in a hospital at an alternative location. In all cases, access to healthcare information might be required in real-time.

Each stakeholder has a different duty to improve the provided services. For instance, Doctor X should be able to access heart rate data, whereas Doctor Y might need to monitor the blood pressure of the same patient; parts of the data might be used by an insurance company for further services. The CU should be configured and assembled in such a way to allow for varying requirements for access policies and enable the patient to control who has access to what information. In addition, for any outdoor open environment, the patient’s CU must be able to change the existing policy through their own access level if needed in an emergency situation [69]. Hence, the patient’s CU can manage and control the accessibility of healthcare data based on changing circumstances.

Based on these scenarios, the roles, activities, and duties of a user play a very important role. Therefore, it is recommended that a Role-Based Access Control (RBAC) [70] policy should be adapted. RBAC needs to be combined with the privacy policy to identify the internal and external conditions required for any stakeholder to meet the specific access control requirements. To do this and achieve the requirements of the access policy and the different roles in a WBAN system, we need to provide different scenarios, such as centralised and decentralised networks, for individual or groups of WBANs to transfer sensitive data on both a small and large scale. The most general requirements are a high level of security and privacy through the use of encryption, decryption, and the protection of related data and access control [71]. An appropriate healthcare architecture is required, but several challenges exist in terms of security and privacy within and between the healthcare domain that need to be addressed.

### 3.2. System Architecture

To develop a data flow model for WBAN system architecture in the health area, both indoor and outdoor environments, some WBAN applications, and related system architectures need to be discussed. Furthermore, the recognition and analysis of different parties in WBANs communication, such as stakeholders, location, and WBAN applications in each scenario, are discussed and presented to demonstrate the relationship between different parties and how data transfer occurs among stakeholders. System architecture in the healthcare environment is divided into three tiers: intra-, inter-, and beyond-WBAN communication, as shown in Figure 4. Communication between different nodes in WBAN, such as the CU and sensor, is one-to-one and one-to-many, which is explained further in this paper. Various technologies, including Bluetooth, 5G, and Narrowband (NB), are used to transmit data among any of the stakeholders in WBAN [10,21,40,72].

#### 3.2.1. Intra-WBAN Communication (Sensor Networks (Tier 1))

It comprises several sensors attached to or implanted on the body to monitor physiological signals. The communication in intra-WBAN is radio-based and includes communication between on-body sensors, and communication between on-body sensors and external devices such as personal/smart devices, laptops, and computers, and a variety of access points. Intra-WBAN communication is a core component in WBAN communication because the entire WBAN depends on it: the personal data are forwarded from intra-WBAN communication to inter-WBAN communication for processing [73,74]. Security, access, and data flow among sensors in intra-WBAN and connections between sensors and the next tier are critical issues that will be considered in Section 5.

#### 3.2.2. Inter-WBAN Communication (Mobile Computing Networks (Tier 2))

Inter-WBAN communication flows between personal devices, environmental sensors, and one or more gateways. Devices in this tier are considered to have sufficient resources and capabilities to record and process data over potentially long periods of time [56]. As shown in Figure 4, personal devices may accumulate body sensor data and forward data to different gateways, such as access points that redirect data via the internet to other WBAN networks [7]. The aim of inter-WBAN communication is to connect different networks to transfer personal data more easily. Communication in this tier uses both existing infrastructure and ad hoc connections and network architectures [21,73]. Security, access control, and data flow among mobile computing in intra-WBAN communication and the next tier are critical issues in WBAN systems [75]. It is crucial to prevent unauthorised access to medical resources and secure information flow and routes in WBAN systems. In addition, there must be an authorised third party from the third tier to transmit data securely.

#### 3.2.3. Beyond-WBAN Communication (Backbone Networks (Tier 3))

The last tier of system architecture is the back-end of WBAN systems, which includes various kinds of nodes and systems connected to WBANs over internet infrastructure, including medical professionals, emergency services, and individual patients (Figure 4). This tier provides different services and interfaces to access data and transmit it to other networks and locations such as hospitals, homes, and databases. Beyond-WBAN communication generally improves the application of WBANs in healthcare, for instance, by enabling physicians and emergency teams to access vital patient information anywhere and in real-time. Furthermore, storage and database management in beyond-WBAN communication is essential to enable extended healthcare data processing.

Beyond-WBAN communication is able to improve the coverage range as well as the application of remote healthcare services. The structure of communication in this part depends on healthcare service provider requirements in WBAN. The security and privacy of data are also critical, just as in inter-WBAN and intra-WBAN [7]. Communication and access requests between users from different domains increase security and privacy issues as each healthcare domain has different settings. Hence, each domain authority includes entities with various different security and privacy settings required to prevent against malicious and unauthorised access to healthcare data by insiders and outsiders.

### 3.3. Communication Technologies

The WBAN system and different communication between stakeholders and environments are presented in a hierarchical system by inter- and intra- and beyond-WBAN communication in Figure 4. The CU communicates with an inter- and beyond-WBAN to address any requests. Different devices, such as an access point or phone, redirect data to the third tier [76], where data management occurs between authorised users. According to the developed data flow model and architecture of the WBAN communication, we consider indoor and outdoor WBAN communication in this study. To transfer data between sensors, CUs, and other devices, as well as between physicians, a point-to-point (P-P) [77] method is used at the first tier and one-to-many communication is considered at the second and third tier [6].

Technologies such as Bluetooth, NB, and Wifi accommodate different requirements in inter-, intra-, and beyond-communications. Therefore, a short range; low power; and low-data-rate radio communication are needed to support wireless communication in e-healthcare systems. Additionally, power consumption is critical to improve the life of sensors during communication [78]. Therefore, performance measures to support WBAN applications include wireless technology enabled with low power consumption; latency; frequency; and a data rate with a long lifetime to optimise the efficiency and reliability of e-healthcare systems [40,79].

In this architecture, sensors capture physiological data and forward it to the second and third tier to be monitored and used by physicians. Single-hop and multi-hop communication are adopted to transfer data among devices and stakeholders in the developed models. The routing protocol [80,81] also plays an important role in the quality of transmission among WBAN tiers, satisfying WBAN requirements like energy consumption, delay, and network life time. To achieve a reliable routing protocol, QoS [80,82,83], cross layer, and cluster-based routing protocols are developed [84]. Existing routing protocols in WBAN communication can be considered for the developed WBAN communication model but will require modification to these protocol/s or the development of new protocols to cover all the requirements [85].

### 3.4. Standards in WBANs

The consumer electronics industry has rapidly moved current standardised wireless technologies for industrial automation, including IEEE 802.15.6 and Industrial, Scientific, and Medical (ISM) radio-based protocols for personal area networks, to meet the needs of WBAN [86,87,88]. The latest wireless standards and technologies concentrate on network construction related to short-range communication, and low power and cost to satisfy the minimum requirements of a wireless technology in healthcare systems in terms of implementation [40,89]. There are many different available wireless technology standards with different focuses. As an example, IEEE 802.11 focuses on high-speed communication, 802.15.1 focuses on personal area networks, and 802.15.4 focuses on close wireless communication as well as low power consumption. The latest standard is IEEE 802.15.16, focusing on WBAN and relative applications [35,90,91,92], which was established to raise interoperability between all medical devices, both industrial in and on the human body. TG6 is an emerging technology in the industrial wireless standard for low-data-rate, highly reliable, and low-power devices (Figure 5), with possible industrial applications such as control and personal healthcare systems. Another focus of this standard is to define the physical (PHY) and medium access control (MAC) layers to provide high-quality communication for medical devices in terms of low power [93]. This standard covers a variety of frequency bandwidths such as UWB, which helps to provide different levels of security in terms of authentication and encryption [94,95].

### 3.5. Electronic Healthcare Services Overview

We present an electronic healthcare services overview diagram based on the earlier general healthcare scenario, system architecture, and WBAN communication technologies (depicted in Figure 6). There is no existing electronic healthcare architecture that provides an overall view of our review regarding WBANs and healthcare. As presented and explained in Section 2, medical sensors are designed to collect and process healthcare data for further services. Hence, healthcare data are stored and communicated within and between electronic healthcare components as depicted in Figure 6, which poses several challenges in terms of security, privacy, and trust. In healthcare scenarios, a patient is the owner of his/her data and is able to share sensitive data or delegate authorization with any healthcare service provider. An access control scheme is one of the best approaches to control who has access to data based on their duties and responsibilities. Key management is important, both for the access control approach and communication within and between medical components in terms of how securely and accurately they generate the secret key for encryption and decryption in the healthcare environment [96]. In the following, our focus is to provide a different solution and discussion based on access control, key management, and trust and to identify future research opportunities.

## 4. Access Control, Key Management, and Trust Model of WBAN Systems

The WBAN systems which we discuss can be neatly abstracted as connected multi-tiered sub-systems as shown in Figure 4. The abstraction allows for a structured and separated discussion of access control, key management, and trust for connectivity and classification via tiers, respectively.

### 4.1. Access Control

Access control is critical for preventing unauthorised access to healthcare data, either on a medical device or in a database. According to the WBAN system model for communication and technologies (Figure 2), an efficient and scalable access control scheme is required. Moreover, access control must be flexible and lightweight enough for quick updates based on the security setting within a single and between different WBAN systems. This is crucial to provide authorised users access to the required data in different scenarios.

In practice, a patient’s sensitive information is shared within and between domains (e.g., GP’s office or hospital), and therefore professional staff must be able to access the information as necessary. The scheme needs to generate the proper permissions with different privileges, thus granting professional staff access to relevant healthcare data while enforcing different privileges for a different set of users. The domain authority or administrator of the system model must define various policies and permissions that must be suitable and adequate to a particular domain’s security and privacy settings. A user must have access to certain data if the user’s permissions satisfy predefined policies by a domain authority.

Access control must work with WBAN technologies (e.g., wireless channels) alone and must never be a hindrance in emergency scenarios. Unlike current access control protocols for a WBAN, we need to propose a suitable scheme to support different types of networks such as “close-range communication”, as shown in Figure 1. Additionally, each medical device belongs to a WBAN and transmits the aggregated data to the CU via the preferred or available communication channel. The proposed protocols must permit an emergency service/healthcare provider to be authenticated at the sensor or CU under a specified access policy. Lastly, in a real-world environment, a high level of security and access control between devices in a network (e.g., ECC, ECG, and CU) will be required to monitor and enforce individual permissions and privileges for different users and data.

### 4.2. Key Management

E-healthcare applications are enabled by several types of industrial and medical sensors (temperature sensors or ECGs, EMGs, and so on) in a WBAN. Currently, WBAN technologies focus on close wireless communication and low power consumption, although a healthcare WBAN would also include a CU, such as a smartphone, which can aggregate and disseminate data outside of a WBAN. This kind of beyond- or inter-WBAN communication poses integrity, confidentiality, and privacy issues. Fortunately, these issues are not novel and have been solved in other domains. Furthermore, each device involved in inter-, intra-, and beyond-WBAN communication (Figure 4) relies on a relatively capable controller to interface with users and disseminate data. This enables several methods and techniques, which can be used, for instance, to generate and control cryptographic keys in inter-, intra-, and beyond-WBAN communication. To provide a high level of security and to mitigate common issues, the key management protocol (e.g., one-to-one, one-to-many, and many-to-many communication) is one of the cryptographic solutions that can be used. Note that key generation, refreshing, agreement, distribution, and revocation in a WBAN are not straightforward because of resource limitations [77,97].

As a result of the nature of intra- and beyond-WBAN technologies, WBAN technologies face several potential security and key management issues as each WBAN device comes from different security providers with a special secret key [98]. It is impossible to use the keys on the WBAN devices due to security difficulties and behaviour of each company. Additionally, it is not at all easy to adapt the protocols (e.g., Public Key Infrastructure (PKI) technology) on each WBAN device to another device [99]. In other ways, energy efficiency and robustness are prominent and need to be carefully considered in the WBAN system due to resource limitations [100]. Also, to provide better and faster communication between WBAN devices, the transmission power must be high enough, which requires more power consumption. All of this fully demonstrates that the deployment and implementation of such security protocols in the WBAN system are not easy and that many challenges exist in terms of key management [101]. Recently, there have been significant research efforts to overcome these issues, such as proposing an off-body channel model [74,102] and an indoor WBAN technique to solve such issues (refer to Section 5.2 [102]. The authors of [102] presented an indoor WBAN technique to solve such issues, but it does not support on-body communications (refer to Section 5.2 for more examples).

### 4.3. Trust

The Trusted Computing Group informally states that “an entity can be trusted if it always behaves in the expected manner for the intended purpose” (https://trustedcomputinggroup.org, accessed on 14 March 2023). Trust is a central requirement in WBANs and is leveraged to increase customer acceptance. Overall, trustworthy products need fewer security patches and are generally seen as more reliable [103]. WBANs, that is, the networks including devices, are expected to be highly trusted and trustworthy due to their sensitive application. If we can assert that individual elements of a WBAN system can be trusted, we may be able to produce a trusted WBAN system. Currently, we observe that a WBAN system is a highly connected mixed security environment. Each tier in a WBAN system architecture presents different challenges—some of these challenges are considered solved and some are longstanding and open issues.

Tier 1 is composed of mixed category devices that generally are not very powerful and potentially heavily resource-constrained. The more constrained devices in a WBAN system are at the same time important for sensing and actuating, especially when they are implanted. Because of their critical nature, sensors and actuators must be absolutely trustworthy [104,105]. Other devices in the same tier are CUs, which may be used for collecting and possessing sensor data similar to current edge computing approaches [106,107,108]. In a tele-medicine scenario, CUs act as the supervisors of critical sensors and actuators. Their input and output may be used to inform and adjust treatments and implement adjustments by manipulating actuators. Because of their critical role, CUs also must be trusted in our WBAN scenario. Furthermore, trusted CUs have the potential to perform important tasks beyond collecting, protecting, and sharing data, such as implementing rudimentary monitoring capabilities to check the security state of some connected devices, thus implementing important security monitoring capabilities [109].

Tier 2 encompasses a class of devices that are typically consumer-grade, such as smartphones, wearables, and personal computers. As described in the previous sections, devices in this tier are supposed to act as relays with internet connections between healthcare systems (or other systems) and beyond WBANs. Beyond relaying information, these devices are supposed to act at most as temporary storage devices (e.g., to offload and upload CU data). Devices in this tier have to be looked at in a differentiated way: since they are controlled and operated by the owner, they must be regarded as untrusted unless the owners are trusted or some other mechanism has been used to establish their trust [110]. However, such devices have the ability to improve security and thus increase trust in the system considerably. Modern devices already offer a plethora of security features and are able to receive frequent updates and security patches. Some manufacturers even implement trusted execution environments, which may be used for confidential data processing on the device [111]. However, even with the implementation of trusted execution, trusted secure, or confidential computing techniques and technologies and despite the research attention such everyday devices attract, it is still difficult to fully establish trust in them [112,113] and utilise them in a trusted process in the presence of known security vulnerabilities [114].

Unlike Tier 2 devices, the devices and systems of Tier 3 are required to be owned and operated exclusively by trusted parties. The systems are best described as cloud-based, with terminals and access for patients, healthcare centers, and emergency personal. Although we assume that the system operators are trusted, the variety and complexity of the devices and systems involved mean that they must at least be considered as susceptible to compromise. In a trust model, this setup can be summarised by treating infrastructure and devices in the third tier as semi-trusted [115]. Figure 7 presents a high-level view of interactions in a WBAN system (outlined in Figure 2). An important observation is that WBANs are interconnected with important services in Tier 3 using CUs. Additionally, CUs may need to communicate with Tier 3 devices through untrusted Tier 2 devices and systems.

### 4.4. WBAN Security Threats

Based on the presented healthcare services model (Figure 6) and adhering to the compatibility requirements of WBAN, we describe and determine the security threats and attacks related to WBANs. According to the nature of medical hardware that is used for communicating data on a wireless medium, some threats pose immense danger to the hardware. Hence, to deploy BANs in health, we need to consider security and relevant threats and attacks to protect data and prevent unauthorised access to medical data. We identify the most suitable security requirements to protect private data from any adversarial attack. The healthcare service model shows a security scenario of healthcare data communication based on the data flow model in this study. It indicates how medical data flow between different stakeholders, such as patients and physicians, and how an adversary can affect the system.

Several past researchers worked on different types of attacks on individual WBAN communication, which tracks the communication between devices in intra-WBAN communication. Mostly, attacks are not considered at the initial phase of the communication, and no particular attacks on CUs have yet been reported. In the healthcare domain, a passive attack can potentially break the privacy of patients and can be potentially life threatening [116]. A variety of active attacks in the healthcare domain depend on attackers capabilities, for example, monitoring communication in WBAN [116,117]. Different types of passive and active attacks attempt to look into critical data in WBAN communications.

 *(1)* 
*Attacks in wireless communication:*


The eavesdropping and monitoring of data in a WBAN occurs more than any other threat in wireless technology [118]. In this threat, the adversary can eavesdrop on the communication between two parties and entrap the entity during transmission. Interception and message modification are two important threats in medical environments, posing a great security concern as they actively work to remove, modify, and inject false personal data into the medical environment. According to the responsibility of each stakeholder, an attacker can capture data and modify it. This forging permits other stakeholders from potentially administering medicines, which may be dangerous for patients.

A routing attack can create an incorrect path for packets in a network that the adversary can use to detect and capture data during the communication or transmit it to other nodes. It can also change or modify the information, all of which is dangerous for the WBAN system. Denial-of-Service (DoS) is another harmful attack in the healthcare environment because different applications in the BAN system monitor the state of patients in real-time [119]. Strong user authentication is an important security requirement in this field since the field’s radio communication abilities are particularly susceptible to unauthorised internal and external access [120,121]. Mutual authentication is a major problem for real-time monitoring in healthcare environments [122], and it is important to ensure that both entities are authenticated before communicating with each other [123]. Based on different queries and responses from physicians, it is important to ensure that data are updated (often termed in literature as *data freshness*). Unforgeability is an important security requirement that is able to prevent the masquerade attack in WBAN communication [124].

Based on the developed model, a number of sensor devices capture physiological signals and transfer them to a healthcare service provider to improve the quality of services being offered. It is significant to use fewer key pairs in the network to secure data transmission in the WBAN system. Therefore, secure communication with high nodes and a few keys in the WBAN system is needed to reduce the storage overhead in the WBAN system. In addition, the message size is a critical element in reducing the storage overhead and energy consumption for inter-WBAN and intra-WBAN communication in the WBAN system.

 *(2)* 
*Attacks in WBAN application:*


In addition, based on the presented electronic healthcare service model and motivation of addressing the challenges discussed above, external users such as physicians may need to access medical resources if they need medical information related to the patient. This demand can lead to several types of vulnerabilities arising in WBAN communication. According to initial authorization between sensors and the CU, the adversary can access medical data that were previously recorded by the CU. Also, the CU can forward a query to the medical server or physician for further services, creating a loop hole for the entire communication system. Here, the adversary is able to communicate with other nodes just by having access to the shared secret key. This scenario is applicable in the case of an adversary attacking communication from a physician to access medical resources as well. It is therefore necessary to apply access control based on a variety of roles, policies, and access levels to mitigate this particular type of attack and prevent unauthorised access.

 *(3)* 
*Attacks in WBAN devices:*


In the case of multiple physician access like doctors and nurses, from either the same or different locations, an appropriate level of access in terms role and policy is also developed [69]. We assume that an adversary is able to attack devices that are held by the physician (e.g., PDA, laptop). These devices communicate with other base stations to monitor data in real-time and on demand. As a result, an adversary can carry out a wide variety of attacks that may endanger critical data from different medical databases (e.g., PDA, sensor, medical server and home server). Based on this, an adversary is able to access all medical data in any place as the doctor has full permission to access medical resources. To recognise and prevent this type of attack, it is important to prepare a strong security mechanism to satisfy security requirements and user access control (user authorization, authentication, and accountability). It is important to know which stakeholders have the ability to read and share data. As mentioned in the developed model, every patient carries different types of sensors; thus, adversaries are able to attack these sensors for forgery, injection, reply, and modification. While this attack occurs at the initial network deployment phase, the adversary is able to obtain a shared secret key and communicate with CUs held by patients. These attackers are able to access recorded medical data or create false data that can pose threats to the life of the patients.

## 5. Discussion, Open Issues, and Future Research Direction

In this section, we highlight and discuss the main security, privacy and trust challenges for WBAN that are appropriate for the presented WBAN system model. We provide recommendations and current research opportunities that can meet the access control, key management, trust, and database management challenges for further studies in a WBAN system. In general, the main target of this section is to provide insight into the future research trends and direction in WBANs.

### 5.1. Access Control

#### 5.1.1. Access Control Problem and Issue in WBAN

We investigate the presented WBAN system model based on access control techniques for the healthcare environment. The privacy of medical data is an important issue that depends on an appropriate access level for each stakeholder. A variety of roles, responsibilities, and levels of access are assigned to different stakeholders, which allow the stakeholders to remotely monitor healthcare data. This management prevents the privacy of information being disputed.

Over the past decades, many access control approaches have been proposed to improve security and privacy. The most essential access control models are the access control list (ACL), mandatory access control (MAC), discretionary access control (DAC), role-based access control (RBAC) [93], and attribute-based access control (ABAC) [125,126,127]. To provide MAC models in WBAN, a Fuzzy logic scheme has been used in WBANs [128]. However, most ACL, MAC, and DAC models are not suitable for the CU and sensor levels for network performance reasons. According to the presented WBAN system and generic healthcare scenario, RBAC is a good choice in WBAN communication. It can be used as the most convincing access management methodology to allow stakeholders to manage data or resources. Unfortunately, RBAC is not scalable and involves minimal complexity for secure WBAN communication. It proves to be inappropriate when role assignment needs to be updated frequently in a large group of users. The authors of [129] proposed an access control model using the concept of RBAC and data provenance models. However, such a model is not suitable for WBAN. To address this problem, studies proposed models such as the Context-Aware Access Control (CAAC) [130] and Critical-Aware Access Control [131]. However, these models cannot completely eliminate access collisions.

Ref. [132] suggested a lightweight and secure ABAC model using the concept of a *signcryption* scheme, which is a combination of encryption and a signature scheme. Although this type of scheme is secure, it is not well suited due to the current limits of medical sensors. Identity-based authentication is proposed in [133] to provide access control within and between WBANs systems, using advantages of a key agreement scheme. Although useful, the model increases communication and computation loads. A Dynamic Cross-Domain Access Control Model for Collaborative Healthcare Applications has been proposed to distribute the attribute in multi-domain [134]. However, the proposed model lacks invocation. Ref. [135] proposed access control using an encrypted identity-based signature (IBS) scheme to address the lightweight medical sensors’ low power consumption and storage capabilities in WBANs. However, this type of scheme might not be suitable for WBANs because of its signature size.

So far, a number of asymmetric key cryptography studies have been carried out [136]. These methods are associated with Elliptic Curve Cryptography (ECC), to provide better security and give correct access control [137,138]. Thus, healthcare service providers can establish a pairwise key with the patient devices to encrypt and decrypt the data for transmission. Later, [139] presented a certificate-based access control and key agreement scheme using the Hyper Elliptic Curve Cryptography (HECC) concept with a one-way hash function. However, providing the appropriate access level using symmetric key cryptography is difficult because of the high design complexity and computation time associated with pairwise and group-wise key management mechanisms. In addition, the adversary is also able to access keys, resulting in a node being compromised.

More recently, researchers used public key cryptography methods to provide better access control as encryption in public key cryptography is based on one-to-many associations needing less communication and computation in WBAN communication [58,140,141,142,143,144,145]. Attribute-Based Encryption (ABE) is a well-known method [146,147] and is used to provide better access control with less key management complexity in WBAN [148]. Ref. [149] proposed a scheme to secure the data communications with and between medical sensors in WBAN and the CU by using the concept of a Ciphertext-Policy and signature. In this scheme, the communication and computation overhead increase because of the concept of ABE and signature. ABE is also compatible with the RBAC [93] and CAAC [150] mechanisms, which help to provide a desirable access level with less key management complexity in WBAN.

Access control is an important aspect of healthcare and the WBAN system. We presented a comprehensive review of WBAN and the healthcare system, mainly based on WBAN applications, communications, and architecture. Open challenges inspired by access control features are:Which access control model is most accurate and suitable for the WBAN system in the healthcare environment?Which access control model can dynamically be adapted with centralised and/or decentralised healthcare system?

#### 5.1.2. Future Direction for Access Control in WBAN

Providing an adequate level of access control is another serious security property to be addressed as unauthorised access to sensitive data can break patient privacy. It is important to provide an access control model with the least complexity to address the limitations in WBAN resources, such as data storage and energy consumption. Although a number of access control models have been proposed for intra-WBAN communication, such as ABE and symmetric key cryptographic models, these models are complex and unable to satisfy the security property of WBANs. Hence, it is useful to adopt an access control model using the concept of access control policy setting. In addition, it would be better to provide an access control based on the future direction of wireless channels as this will enable a system to meet the security requirements without using additional equipment.

### 5.2. Key Management Issues and Open Problems

We describe the most popular key management solutions in WBAN systems: traditional, biometric, and wireless channels. We also briefly discuss cryptographic agility and the need to move to post-quantum cryptography.

 *(1)* 
*Traditional key management schemes:*


There are several traditional schemes [151,152,153] proposing pairwise and groupwise key management protocols to generate and distribute the secret key within and between WBAN systems. These approaches use different key exchange protocols such as Diffie–Hellman (DH) and Elliptic Curve Diffie–Hellman (ECDH). For example, the KALwEN scheme was proposed to establish a secure channel between two sensors in BSNs. In this, a Faraday-cage (FC) is used to manage the pre-distributed keys within sensors. This mechanism introduces extra cost to the BSNs due to new hardware deployment, requiring more communication and computation [154]. Public-key cryptography (PKC) can also be used to set up symmetric keys within and between sensors. Thus, novel key management schemes based on ECDH key generation are used within the medical sensors. The proposed schemes take less than 6 s to authenticate the certificate; however, this algorithm is not suitable for sensor networks [155]. Similarly, a PKC authentication scheme has been proposed to control communication between each sensor and smartphone in a WBAN system. Although the proposed key management schemes are easy to implement within the WBANs systems, the total computational time to generate a key is too long. The scheme is not secure enough due to using a pre-distributed key available in the third-party company [156].

BARI+ is another key management protocol proposed to secure point-to-point communication within and between sensor nodes and medical servers. To generate and manage the keys, BARI+ uses physiological values and is added to a key refreshment schedule. This helps to manage and control the keys but requires extra communication and computation within and between medical sensors and medical servers. Additionally, this requires a third party authority to validate the certificates between domains [157]. A distributed group key agreement protocol has been proposed to secure communication in BANs. The initial key in this scheme generates based on the sensor’s ID, which entirely relies on a PKC to establish a secure channel between nodes. Although the proposed model is high-speed in terms of creation and managing the keys, this model is not suitable for sensor networks due to minimal computing resources available on sensors [152]. To address the issue of key management in the one-to-one communication within sensors, Ref. [152] proposed BANGZKP. However, this model is non-compliant with postural moves, which is unsuitable for WAN sensors.

These protocols are not suitable for medical devices due to their high computation and communication overheads. For example, to address the key management in WBAN, the authors of [154] used the Faraday Cage model [158], which is employed as a secure channel where the communication is unicast from system to sensors. The main idea is to pre-distribute all keys to body sensing devices before deployment. However, to add a new sensor, it is required to run the Faraday Cage to broadcast new key material for the new nodes and refresh the existing keys. Another traditional cryptographic scheme that can fulfill the minimum WBAN security requirements is using the ECC concept [159]. Using this and the advantages gained by using radio-frequency identification tags is one of the best candidate schemes in practice, but this method is not working well in a dynamic system. Although the advantage of such schemes is to enable secure communication between nodes, this types of model incurs high cost and in the most WBAN real scenario is impractical. Not only are using traditional cryptographic approaches in WBAN not suitable, but these approaches also do not meet the limitation of sensor devices and relative networks in which healthcare data flow. Ref. [6] presented a hybrid device authentication model to overcome inter-WBAN communication, but the proposed model cannot support many to many communication simultaneously.

Recently, Ref. [160] proposed a WBAN authentication protocol using the ECC concept. However, several security loopholes are discovered through cryptanalysis of the proposed protocol. This work was improved by [161] to overcome the security vulnerabilities, but the presented result showed the authors failed to achieve this. To overcome these issues, Ref. [98] proposed an identity-based anonymous authentication and key agreement protocol for WBAN. The authors proved that this scheme achieves mutual authentication and user anonymity. However, it does not achieve proper anonymity within and between WBAN sensor networks because managing digital signatures for each device is highly inefficient. Ref. [162] presented a certificate-less online/offline signature and an authentication technique for WBANs, but there is no performance and security evaluation yet.

 *(2)* 
*Biometric key management schemes:*


A general method for securing WBAN communications is using biometric approaches. These types of approaches involve using cryptographic keys to control and handle sensor devices, where each of the sensor devices is able to authenticate the user using biometric feature. A number of biometric methods have been developed to address the key management problem. There are a number of approaches [29,163] proposing pairwise and group-wise key management to generate secret keys and later use those keys for data management (encryption and decryption). Although the outcomes of current studies demonstrated that using the advantages of biometrics is very helpful for different aspects, such as the identification of and ability to provide the needed secure properties, this requires additional hardware, which is not practical for sensors. Additional hardware has high cost and requires more energy for computation, which may not be suitable due to the limitations of medical devices [164]. For example, Ref. [165] proposed Ordered-Physiological-Feature-based Key Agreement (OPFKA), which works similarly to the fuzzy vault approach. A fuzzy valet permits a secret key to hide using diverse key values. The valet can be unlocked if the second set of key values has a similar set of values. The set of values is based on physiological signals. Thus, once senders and receivers obtain both values, the OPFKA scheme runs to generate and manage the keys between both parties. The authors of [166] investigated the security of the current proposed model using biometric schemes on Implantable Medical Devices (IMDs). The most important finding of this article is that using biometric schemes is not acceptable within sensor networks in BANs. Another fuzzy vault approach proposed in [167] uses the frequency-domain features of photoplethysmogram (PPG) to measure the errors between and within medical sensors in BANs. The proposed model has been prototyped to show the system’s feasibility; however, the security of such a proposed scheme is not enough because of the future of the size of PPG. Recently, Ref. [168] proposed a novel attack technique called the Synthetic Electrocardiogram Attack Method (SEAM) to improve a key management issue between WBAN devices. However, SEAM relies on the use of biometrics stored in ECG, and primitive relies on multiple biometrics to enhance the key generation process.

 *(3)* 
*Wireless channel key management:*


To overcome the existing problems, researchers have focused on physical-layer security such as advanced hardware, out of band communication channels, and wireless channel measurements based on key generation, which help to provide better key establishment between medical devices and meeting the limitation of sensors. However, advanced hardware approaches require more energy, and out of band communication channels are sometimes not available. With regard to existing literature [91,101,158,169,170,171,172,173,174,175,176] and to meet the most recent wireless technologies in WBAN for industrial automation, wireless channel property (Received-Signal-Strength Indicator(RSSI)) is a promising technique to generate a secret key between and within WBAN communications. RSSI measurement between two devices can be used as a source of common randomness in wireless communication, and in terms of the position and motion of body, the channel cannot be guessed (considering the de facto security example used) with eavesdropping in another location. The authors of [158] proposed secret key generation (pairwise) in WBAN using RSSI. Ref. [101] proposed an adaptive network that learns from wireless channels and claimed that this reduced the overhead problem, although it does not address the key management issue. Recently, a novel lightweight authentication was proposed by [98] to provide anonymity; however, this work cannot support high rate wireless channels in WBAN system from our analysis. These are almost all theoretical, and the result of their approaches is high rate keys with high bit mismatch rate. Additionally, there is no such work in physical-layer security based on wireless channel measurements to generate group key in WBAN communication, which can be a future research topic.

We identified two major open challenges regarding the key management in the WBAN system:How can wireless channels or biometrics be used to generate a key pair in a WBAN system?How can one propose an efficient key management protocol to meet the security requirements of WBAN and the healthcare system?

#### 5.2.1. Cryptographic Agility and Post-Quantum Cryptography

WBANs, just like any highly interconnected digital system, face the issue of outdated cryptography. The promise of quantum computing only highlights this issue as quite a number of things can lead to compromised cryptography and security in WBANs. Quantum computing in particular has been cited in recent years and even decades as a massive disruption for cryptography [177]. In theory, quantum computers will be able to efficiently solve problems like prime factorization, a key mathematical concept in widely used public-key encryption schemes. However, it is unclear how this will be scaled up, in a fault-tolerant manner, to modern key-sizes of 4096-bit in practice. The authors believe that quantum computing plays an important role in highlighting the need to implement cryptographic agility into WBAN devices, especially those with a potentially long service life. Cryptography can become compromised in a number of ways, e.g., algorithms and mathematical puzzles may lose their complexity because of advances in computing and computing power or because of flaws in widely used algorithms; their implementation may be discovered and practically exploited; and key material may be leaked and become compromised.

Cryptographic agility [178] is particularly crucial in the context of long-lived systems—those potentially used WBANs and the healthcare sector in particular. These systems may need to remain secure over many years, and the ability to adapt to changing cryptographic requirements (updated algorithms, keys, or key certificates) is essential for maintaining a desired level of security.

#### 5.2.2. Future Direction for Key Management Issues in WBAN

To the best of the authors’ knowledge, RSSI is regarded as the best candidate to generate secret parameters in intra-WBAN communication because it is readily available without any computational inclusions or additional hardware and communication. Generating the key secret pair using the future of wireless channels is very useful as this enables the system to achieve a perfect secret key in intra-WBAN communication. Not only does this enable the key generation protocol to generate unique keys based on the future of wireless channels but it also prevents a third party from generating a similar secret key pair. Although there are several peer-to-peer and dependable key generation schemes that have been proposed in WBANs, group key management based on the future of wireless channels is in its infancy.

### 5.3. Trust Issues

The type of trust discussed in the literature is strictly in the sense of device operation, as well as adherence to a specification and the verifiable achievement of its security goals. Another related topic is trust in a social sense, which needs to be considered as well as general Quality of Service (QoS) requirements [179]. People such as patients or healthcare providers need to trust and feel safe with WBANs. Building such trust will be an important factor in the rapid adoption of WBANs. The security and trustworthiness of devices and systems themselves are important factors in the overall trust users place such technologies. Security and trust in healthcare systems are complex topics concepts. Devices that must be trusted might not be completely secure, e.g., they might have unknown vulnerabilities, and a perfectly secure device might not necessarily be trusted when it is in the hands the wrong party. For example, the sensors connected to a CU might be manufactured by a trusted manufacturer but might still be considered as insecure in the sense that they cannot selectively share data with CUs or provide a unique identification for themselves. In such a case, we must acknowledge that the device is insecure but must be trusted regardless. Similarly, sensor devices in a WBAN must be trusted, and we may have to accept the possibility that, at least for now, we will not be able to establish trust in them. One way to make a WBAN trustable in the near future would be to promote the use of devices coming from a set of trusted manufacturers and maintainers. This can justify some of the implicit trust by referring to the trusted manufacturers and maintainers. If sensors could at least be authenticated, WBAN systems could allow for the mixing of sensors from different manufacturers. Optimally, sensor devices should be trustworthy, and using only trustworthy sensors would also turn a WBAN into a truly open system with verifiable components.

Sensors in a WBAN are connected to CUs, which are loosely coupled with multiple sensors. The authentication of particularly low-powered sensors is a pressing issue and prevents one from properly defining the coupling between sensors and CUs [180]. Even if the intention is to create a trustable WBAN system by using only trusted components, we cannot reliably prove that we have done so because we cannot reliably identify individual components. The issue of only composing trusted components into a trustable WBAN will have to be deferred to manufacturers, maintainers, and ultimately suitable professional staff.

In the WBAN model, CUs are the only and by default also the most important link between sensor data consumers and the sensors itself—a setting that is similar to that of edge computing use-cases. Among mechanisms to potentially react based on some sensor readings, CUs are trusted to accumulate and relay sensor readings to Tier 2 and Tier 3 devices in Figure 4. In the previous decade, CUs were single-purpose built devices. However, as devices have become smarter, many functions that previously required multiple purpose built devices are now being combined into one device. Examples include current smartphones, which offer almost ubiquitous connectivity, relatively large storage, and considerable computing and data processing power. From this perspective, it is reasonable to assume that CUs will be implemented as a software service running on a smart device. Consequently, the CU will run on a commodity (mobile) operating system alongside other functionalities and applications. Historically, medical equipment comprised purpose built devices with constrained interfaces and resources that were generally not modifiable or managed by their owner. Providing trustworthy CU software and ensuring that the CUs context (both the software and hardware) are trustable will be a major challenge in the future. In this area, WBANs will share trust issues similar to those of services relying on security guarantees that must be kept by a client application.

We assume that intra-WBAN communication is mostly an infrastructure issue. However, WBANs may also need to provide immediate access to stored and live data to privileged entities such as emergency services and medical professionals. WBANs may use other WBANs as relays in an ad hoc scenario where dedicated infrastructure is not present. Finally, WBANs will use cloud services to offload, store, and backup otherwise local data and in some cases fetch or retrieve records if they are needed but not locally available. For intra-WBAN communication using a variety infrastructure elements, we can remove a large portion of the involved infrastructure from our model. By requiring secure channels between communicating entities, we can constrain the involved entities and agents that necessarily access the data. Considering secure channels allows us to reduce infrastructure concerns to availability, while data integrity and confidentiality rely on the security of the end-points. Furthermore, using certificates and suitable encryption will become realistic as CUs and Tier 2 devices become increasingly capable [181].

Although we can, with reasonable assumptions, exclude some Tier 2 devices (Figure 4), entities, and agents from our trust discussion, we eventually will have to consider propagating data, storing data, and eventually making them accessible by appropriate parties. Considering appropriate parties is beyond this discussion; an important consideration remains as to how we can meet fundamental data integrity guarantees while still making data accessible and meeting confidentiality requirements. A trusted system will have access to data and even control some of it but must be trusted not to propagate or modify it without authorization. While cryptography offers fine-graded solutions with regard to data access when appropriate, we still have to trust that while data are decrypted or under the control of another only appropriate modifications will be made. A practical example includes systems under the control of a practitioner: medical staff such as doctors will have appropriate permissions, but they will also need at least trustable systems that can guarantee that they will not tamper with or leak data that is accessed by the practitioner. On the other hand, when data are stored in the cloud, a likely and desirable scenario, the cloud systems will have to follow standards and implement mechanisms to make the provided service trustable and optimally trustworthy at all times. In short, the issue of sharing data with and accessing data from Tier 3 devices depends largely on the integrity of the devices involved and poses significant security and trust challenges for cloud services and other endpoints accessing the data. However, cloud services could play an important role in addressing these issues, ensuring access, and enforcing permissions on the hosted data. For instance, data could be requested in an emergency and provided by a cloud hosted service. This shifts trust to the cloud but also simplifies the model as it does not rely on trusted end-points using the data.

We identified two major open challenges regarding trust in WBANs:Implementing CU functionality using commodity devices has considerable benefits and would allow for fast and effective adoption. How can we ensure that CU functionality running on the top of and alongside other untrusted applications is not compromised and the integrity of both services and data is guaranteed?When WBANs move between domains and share data with other devices and agents, how can the device or agent establish trust in the WBAN? Vice versa, when a WBAN interacts with a new domain, how can the WBAN establish trust in the agents and devices of the new domain?

#### Future Direction for Trust Issues in WBAN

Another area of research that needs attention and focus is the area of trusted and trustworthy computing. The most pressing issue is to make sure that sensor devices produce reliable measurements and are robust against a variety of attacks and threats. Having a set of trusted manufacturers producing trustable sensors that provide at least some form of authentication would be an important step when composing them in a WBAN. While the sensors are critically important as the eyes of the WBAN, the CU (used to aggregate and report readings) poses a slightly different but equally pressing challenge. The more open and integrated such a CU is and the more a user can configure it, the more potential threats are introduced into the WBAN system regardless of otherwise good sensor inputs. In short, a trustworthy CU must be able to enforce the use of trusted and trustworthy sensors wherever possible. As its key functionality, the CU must be trusted to maintain the integrity of sensor data and related functions at any time. More generally, as we move further away from WBAN sensors to infrastructure and processing, the issues faced by our system become similar to those of other information systems, especially with regard to data sharing amongst domains, cloud storage, and processing.

### 5.4. Healthcare Database

#### 5.4.1. Healthcare Database Issues in WBAN

Recently, the healthcare database is one of the most important issues as healthcare devices and relative WBAN technologies generate a huge number of data that must be securely stored and available anytime. Cloud-, government- and local (hospital)-based databases are the popular choices that provide a way to record data efficiently. Although these databases address the storage overhead on medical devices such as sensors and reduce the cost of storage, these also increase the complexity of the network and pose several security and privacy issues while data are outsourced and stored in any database. The security and privacy issues related to the databases are integrity, availability, confidentially, information leakage through side channels, unauthorised data access, abuse of storage services, data poisoning, data breaches, data loss, authentication, and reliability while data are outsourced and stored in any databases.

We identified two major open challenges regarding the healthcare database:i.How to design a centralised and/or decentralised healthcare database?ii.How to propose an efficient search engine that can be used in both a semi-trusted and a not-trusted healthcare server?

#### 5.4.2. Future Direction for Healthcare Databases

Based on the security properties and system requirements of public and private healthcare databases such as cloud servers, message-dependent encryption, encryption schemes (e.g., symmetric and asymmetric searchable encryption), traffic obfuscation, and deterministic information dispersal are the possible solutions that can be given to secure databases in different forms [182,183]. To the best of the authors’ knowledge, searchable encryption (SE) (e.g., searchable symmetric encryption (SSE) and searchable asymmetric encryption (SAE)) is regarded as one of the best candidates to outscore the data as well as secure the databases against unauthorised users [184].

### 5.5. Summary

WBAN and related technologies have been shown to be useful in the healthcare system. However, WBAN has several issues that must be addressed. Our presented WBAN system architecture and its security implications defined in the previous section highlight that it can be used as a general model for the WBAN healthcare environment. It can also be used in further studies to propose better security protocols addressing the security and privacy issues for remote access to data and resources. We further summarise current research opportunities and provide recommendations in terms of intra-, inter-, and beyond-WBAN communication.

To achieve better security, privacy, and trust in WBANs, we need to propose a lightweight key management approach, and protocols must meet the resource limitations and provide security properties such as data confidentiality, availability, authenticity, integrity, and non-repudiation. Additionally, the development of access control techniques is required to prevent unauthorised access to healthcare resources physically and logically. From a physical point of view, we have to limit and control types of access, such as direct access by manufacturers. From a logical point of view, we have to control and restrict certain users’ actions with the concept of wireless channel properties (e.g., RSSI) and policy settings within and between sensors in WBANs. Finally, we need trusted and trustworthy computing protocols that guarantee that medical devices produce reliable measurements and are robust against a variety of security threats.

We require different levels of security, privacy, and trust to achieve a desirable, efficient and effective scheme in intra-, inter-, and beyond-WBAN communication. For this, we need to pay attention to WBAN limitations in terms of resources and communication, such as power consumption, storage, interface communication, and computation cost. In Table 1, we reviewed, investigated, and compared existing survey works with the results presented in this paper.

## 6. Conclusions

WBAN is an emerging technology that focuses on monitoring physiological data for different applications in the next generation of healthcare monitoring to improve quality of life. In this paper, we presented an overall review of the current state of WBAN systems and relative architecture and communication using different healthcare environments for remote monitoring. The contribution of this research is the analysis of the general WBAN system model for intra-, inter-, and beyond-WBAN communication to identify the future direction of research. We outlined three key challenges in developing a healthcare system, which generally are security, privacy, and trust. We also presented and discussed the current access control, key management, database management, and trust solutions and analyzed these to direct the future research direction in this field. In general, the collected data and results in this paper help familiarise the researchers with the state-of-the-art WBAN and healthcare applications. We believe this work can serve as a source of future work in terms of security, privacy, and trust aspects.

## Figures and Tables

**Figure 1 sensors-23-09856-f001:**
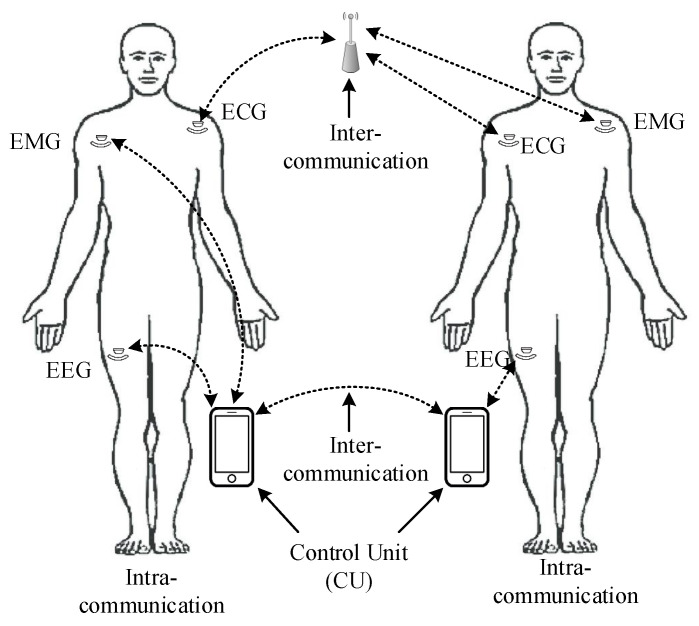
Communication between and within WBANs.

**Figure 2 sensors-23-09856-f002:**
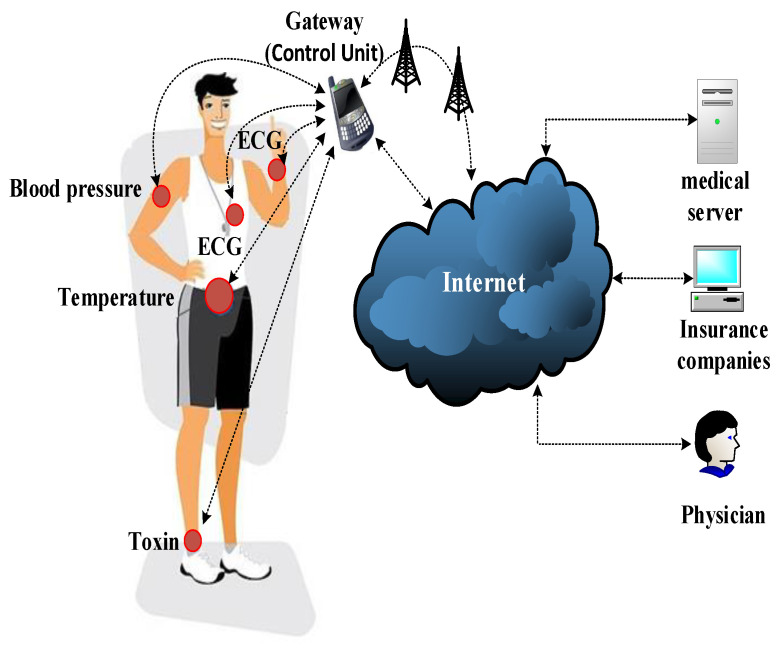
WBANs system model.

**Figure 3 sensors-23-09856-f003:**
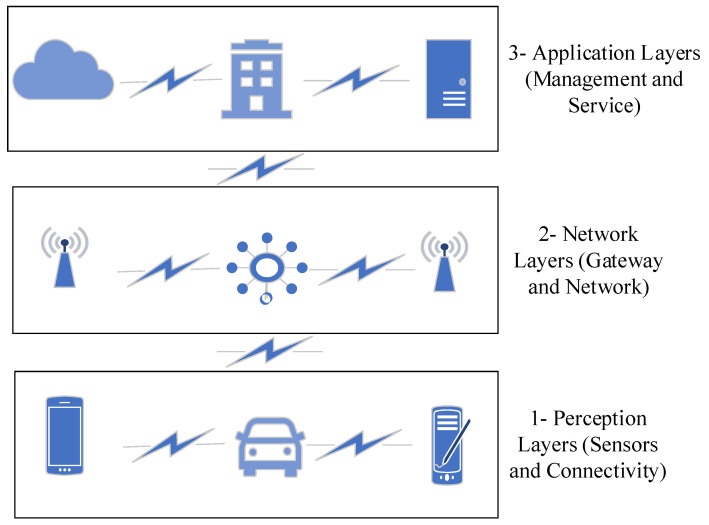
The basic edge computing architecture.

**Figure 4 sensors-23-09856-f004:**
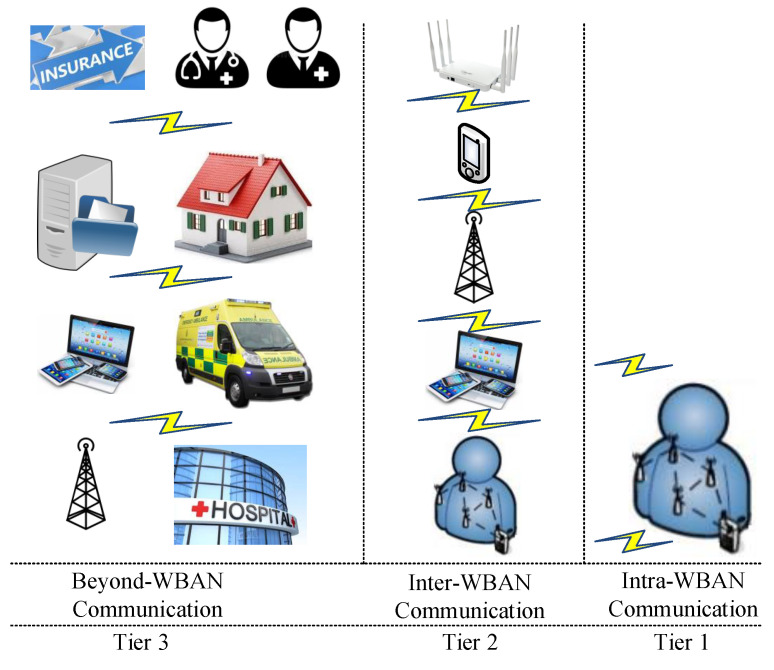
WBAN system architecture in the healthcare system.

**Figure 5 sensors-23-09856-f005:**
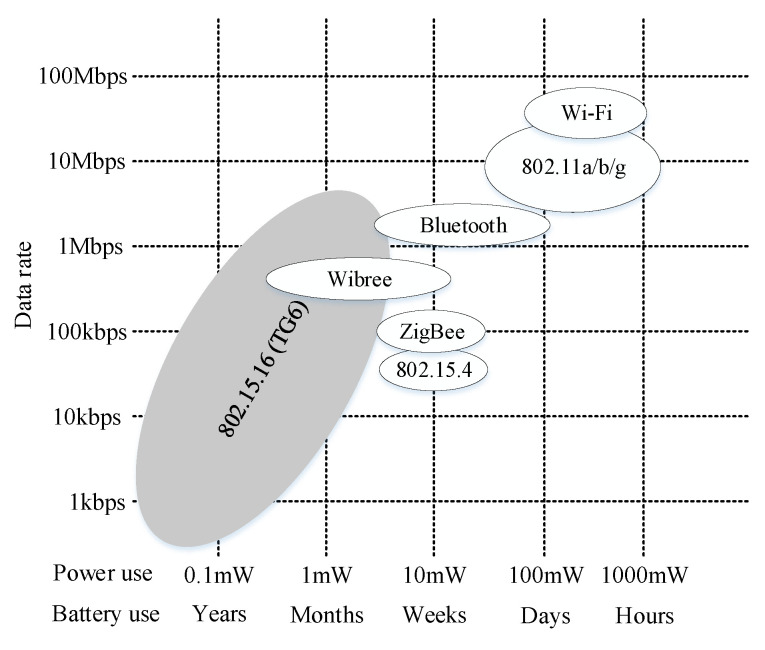
Power, battery, and data rate in the WBAN.

**Figure 6 sensors-23-09856-f006:**
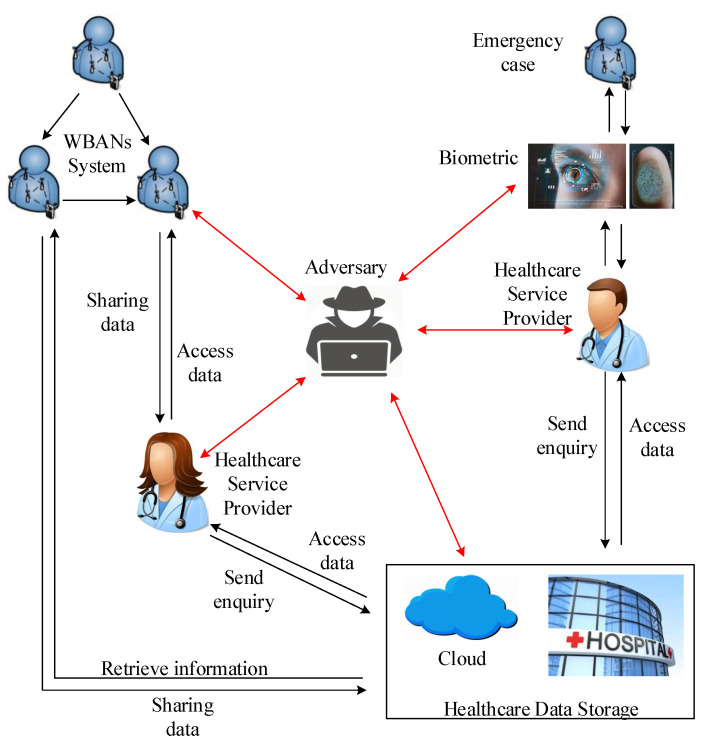
Overview of electronic dataflow in healthcare services.

**Figure 7 sensors-23-09856-f007:**
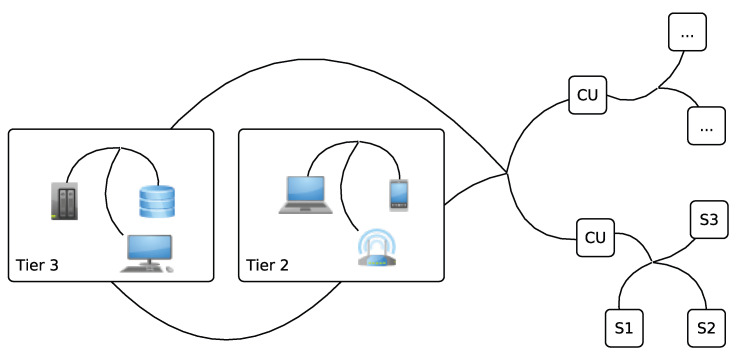
The WBAN system architecture of Figure 2 is abstracted to outline classification in tiers and connectivity. We consider CUs and sensors as trusted, Tier 2 devices as generally untrusted, and Tier 3 devices as semi-trusted.

**Table 1 sensors-23-09856-t001:** Categorization of healthcare WBAN surveys. In this table, PY is the Publication Year, SE is Security, PR is Privacy, AC is Access Control, TR is Trust, OI is Open Issue, and KM is Key Management.

	PY	SE	PR	AC	TR	OI	KM
[26]	2015	**✓**	**✗**	**✗**	**✗**	**✓**	**✗**
[27]	2015	**✓**	**✗**	**✗**	**✗**	**✗**	**✓**
[28]	2016	**✓**	**✗**	**✗**	**✗**	**✓**	**✗**
[31]	2016	**✗**	**✗**	**✗**	**✗**	**✗**	**✗**
[32]	2016	**✗**	**✗**	**✗**	**✗**	**✗**	**✗**
[20]	2017	**✓**	**✗**	**✗**	**✗**	**✓**	**✗**
[23]	2017	**✓**	**✗**	**✗**	**✗**	**✓**	**✗**
[29]	2017	**✓**	**✗**	**✗**	**✓**	**✓**	**✓**
[30]	2017	**✓**	**✓**	**✗**	**✗**	**✓**	**✓**
[33]	2017	**✓**	**✓**	**✗**	**✗**	**✓**	**✗**
[3]	2018	**✗**	**✗**	**✗**	**✗**	**✗**	**✗**
[34]	2018	**✓**	**✓**	**✗**	**✗**	**✗**	**✓**
[35]	2019	**✓**	**✓**	**✗**	**✗**	**✓**	**✗**
[36]	2021	**✓**	**✗**	**✗**	**✗**	**✓**	**✗**
[37]	2021	**✓**	**✓**	**✗**	**✗**	**✓**	**✗**
[38]	2022	**✓**	**✗**	**✗**	**✗**	**✓**	**✗**
[39]	2023	**✓**	**✗**	**✗**	**✗**	**✓**	**✗**
Our paper	–	**✓**	**✓**	**✓**	**✓**	**✓**	**✓**

**Table 2 sensors-23-09856-t002:** Type of sensors in WBANs: ms stands for the milliseconds and kbpa stands for 1 kbit/s (one kilobit per second), 1 Mbpa stands for 1 Mbit/s (one megabit or one million bits per second), VH stands for very height, VL stands for very low, EC stands for Energy Consumption, BER stands for Bit Error Rate (BER), DC stands for Duty Cycle, ST stands for Set Up Times, DR stands for Data Rate.

Sensor	EC	BR	Latency	Bit Rate	DC	QoS	Nodes	ST	Privacy	DR
Accelerometer	Low	$10^{-10}$	<250 ms	<10 kbps	<1%	**✓**	<12	<3 s	High	High
Blood glucose	VL	$10^{-10}$	<250 ms	<1 Mbps	<1%	**✓**	<12	<3 s	High	High
Blood pressure	High	$10^{-10}$	<250 ms	<10 kbps	<1%	**✓**	<12	<3 s	Medium	Low
CO gas sensor	Low	–	<250 ms	<10 kbps	<1%	**✓**	<12	<3 s	High	VL
ECG	–	$10^{-10}$	<250 ms	86.4 kbps	<10%	**✓**	<6	<3 s	–	High
EKG	Low	$10^{-10}$	<250 ms	<192 kbps	<10%	**✓**	<6	<3 s	High	High
EMG	–	$10^{-10}$	<250 ms	<10 kbps	<1%	**✓**	<12	<3 s	High	High
Humidity	–	$10^{-10}$	<250 ms	<250 kbps	<10%	**✓**	–	<3 s	–	VL
Temperature	–	$10^{-10}$	<250 ms	<10 kbps	<1%	**✓**	–	<3 s	–	High
Image	Low	$10^{-3}$	<250 ms	<100 kbps	<50%	**✓**	2	<3 s	Medium	VH
Video	High	$10^{-3}$	<1000 ms	<100 kbps	<50%	**✓**	2	<3 s	Medium	VH
Audio	Low	$10^{-5}$	<100 ms	1 Mbps	<50%	**✗**	3	<3 s	Low	Low

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
