# Peer review of "Access Control, Key Management, and Trust for Emerging Wireless Body Area Networks"

_sensors, 2023, doi:10.3390/s23249856_

Round 1

Reviewer 1 Report

Comments and Suggestions for Authors

The papers offers the review of Wireless Body Area Networks. 

1. The literature sources in the papers are not in the sequence order in the text. Please correct the order.

2. Please compare the survey with the other related ones, why your survey is better? (There are many similar ones)

3. The survey lacks technical details, for example encryption schemes lack technical details, please provide them.

4. Artificial Intelligence Driven Wireless Body Area Networks are not mentioned in the text, it is recommended to add the corresponding section and describe advantages and disadvantages of them

5. Authors write a lot about cryptography, but nothing is mentioned about post-quantum cryptography. There are many works of post-quantum encryption schemes in Wireless Body Area Networks. Please add the corresponding section.

6. The authors say: “Denial-of-Service (DoS) is another harmful attack in the healthcare environment because different applications in the BAN system monitor the state of patients in real-time”. There is not future analysis of DOS/DDOS attack on Wireless Body Area Networks. Please add the mentioned analysis.

7. Please talk about MITM attacks on Wireless Body Area Networks.

8. Conclusion is very poor, please extend it.

Author Response

Dear Reviewer, 

I have attached the response to the comment file for your review. 

Regards,
Ahmad

Reviewer 2 Report

Comments and Suggestions for Authors

The paper is will written and have a positive impact to researchers in field of security, privacy and access control. 
However, the access control mechanism needs to explain with different types point of you like Attributes Based Access Control, Role based Access Control and some others. 
The research directions also needs to be updated and add some additional and important field like blockchain. The I am recommending some articles that will help and suggest to cite.

https://www.sciencedirect.com/science/article/pii/S1389128623004395?via%3Dihub

Author Response

(The authors gave the same response as above.)

Reviewer 3 Report

Comments and Suggestions for Authors

The authors have done a great job of reviewing and presenting a ton of data about Wireless Body Area Networks. At the same time, the specific goal and results of the work are somewhat dissolved in this volume. The most unclear part is the part about trust. It is very strange to consider trust without involving the opinions of patients, without surveys and other sociological studies, it is simply speculative - what it might be like before the Jews. At a minimum, it is necessary to find existing studies of the level of trust and refer to them, since there are no results.

In general, the overwhelming number of described problems with privacy and security are widely known and discussed. Although the authors are indeed very meticulous and provide more detailed data, it is still unclear what the importance of this particular work is. I would like to see some kind of generalization, a diagram that would present all the key parameters, which would allow further researchers to use it to evaluate these technologies.

Author Response

(The authors gave the same response as above.)

Round 2

Reviewer 1 Report

Comments and Suggestions for Authors

The paper is improved.